# Identification and Functional Characterization of CYP4D2 Putatively Associated with β-Cypermethrin Detoxification in *Phortica okadai*

**DOI:** 10.3390/genes13122338

**Published:** 2022-12-11

**Authors:** Lingjun Wang, Hongri Tang, Zhimei Xie, Di Li, Changzhu Yin, Bo Luo, Rong Yan, Wei Sun, Hui Liu

**Affiliations:** 1Department of Parasitology, Zunyi Medical University, Zunyi 563000, China; 2Laboratory of Evolutionary and Functional Genomics, School of Life Sciences, Chongqing University, Chongqing 401331, China; 3Center for Laboratory Teaching of Clinical Skills, Qiannan Medical College for Nationalities, Duyun 558000, China

**Keywords:** *Phortica okadai*, *Thelazia callipaeda*, CYP450, beta-cypermethrin, insect resistance

## Abstract

*Phortica okadai*, a polyphagous pest, serves as a vector for *Thelazia callipaeda* in China. Currently, there are no effective control strategies for this vector. Agricultural pest control may cause *P. okadai* to become a threat due to the development of pesticide resistance. Cytochrome P450s (CYP450) plays a significant role in detoxifying xenobiotics in insects. In this study, we performed RNA sequencing of *P. okadai* exposed to β-cypermethrin for 0 and 1 h and then gene cloning of the five up-regulated CYP450 genes. Three CYP450 genes were successfully cloned, and their expression patterns in different developmental stages and in different tissues were analyzed by RT-qPCR. *Pocyp4d2* was observed to have the highest expression in the midgut (fold change 2.82 for *Pocyp4d2*, 2.62 for *Pocyp49a1*, and 1.77 for *Pocyp28d2*). Functional analysis was carried out according to overexpression in S2 cells from the pfastbac1 vector and RNAi with siRNA. The results of the CCK8 assay indicated that the overexpression of the recombinant protein PoCYP4D2 suppressed the decrease in S2 cell viability due to β-cypermethrin. The expression levels of PoCYP4D2 decreased significantly, and the mortality rates increased from 6.25% to 15.0% at 3 h and from 15.0% to 27.5% at 6 h after *Pocyp4d2*-siRNA injection. These results suggest that *Pocyp4d2* may be an essential key gene in the metabolism of β-cypermethrin in *P. okadai*. This study constitutes a foundation to explore further the functions of *P. okadai* CYP450 genes in insecticide metabolism.

## 1. Introduction

*Phortica okadai* (Drosophilidae, Steganinae) is the only confirmed vector of *Thelazia callipaeda* (Spirurida, Thelaziidae) in China [1], whereas *Phortica oldenbergi* and *Phortica variegate* are known vectors of this nematode in Europe [2,3]. *T. callipaeda*, a zoonotic nematode, can parasitize the eyes of many mammals, such as canids, domestic carnivores (dogs and cats), wild carnivores (red foxes), lagomorphs (brown hares), and humans [4,5]. Over the past two decades, various reports have highlighted *T. callipaeda* infections worldwide, and the alarm has recently been raised regarding *T. callipaeda* infections becoming a significant public health concern [6,7]. Additionally, *Phortica* species are also polyphagous pests, and there are currently no effective strategies for their control [8].

The control of vector-borne diseases (VBD) and agricultural pests has, until recently, largely relied on insecticides [9]. Commonly used pesticides include pyrethroids, carbamates, organochlorines (DDT), and organophosphates [10]. Pyrethroids are a class of synthetic insecticides with low toxicity to mammals and high efficacy against vectors [11] and have generally been widely used. Unfortunately, to enhance their effectiveness, large amounts of chemical pesticides have been used in the past, which has also increased the incidence of insecticide resistance [12]. The efficacy of pyrethroid insecticide treatments has decreased due to the development of insecticide resistance, which has caused pest control efforts and environmental pollution to become ongoing challenges [13].

Insecticide resistance is caused by several mechanisms; to date, four types of resistance mechanisms have been described, including metabolic resistance, target-site resistance, behavioral resistance, and cuticular resistance [14]. Many studies have focused on metabolic resistance in pyrethroid-resistant insect pests. It is known that the increased detoxification through metabolic enzymes is principally associated with three major enzyme groups and the ABC proteins, such as cytochrome P450 monooxygenases (P450s), carboxylesterase (CarEs), glutathione S-transferases (GSTs), and ATP-binding cassette (ABC) transporters [15,16]. P450s, a large superfamily of heme-thiolate enzymes, plays a major role in phase I (biotransformation) detoxification [17]. The increase in the transcription and expression of the CYP4, CYP6, and CYP9 family members enhances the metabolic detoxification of pesticides [18]. In addition, the tissues of resistant insects, including the midgut (MG), fat body (FB), and Malpighian tubules (MTs), are also involved in the metabolism and detoxification of xenobiotics [19,20]. *P. okadai*, a polyphagous pest, is usually distributed around orchards [21]. Given that agricultural pest control may cause *P. okadai* to become a threat due to the development of resistance, understanding the metabolic mechanism of insecticides in *P. okadai*, especially, the role of the p450 enzyme, is crucial for a timely initiation of appropriate integrated pest and vector management interventions.

In this study, three CYP450 genes (*cyp4d2*, *cyp49a1*, and *cyp28d2*) were cloned from *P. okadai*, and their expression patterns in different developmental stages and other tissues were analyzed using qRT-PCR. *Pocyp4d2*, with the highest expression in the midgut, was subjected to functional analysis according to overexpression in S2 cells from the pfastbac1 vector and RNAi with siRNA. This study constitutes a foundation from which to further explore the functions of the CYP450 genes in insecticide metabolism. It may provide a better insight into the improvement of the control management of *P. okadai*.

## 2. Materials and Methods

### 2.1. Insect Rearing

A laboratory colony of *P. okadai*, without exposure to any insecticides during the last 6 years, was established from a field collection from Zunyi, Guizhou Province, China [21]. The insects were reared on fruits (such as apples and pears, with a size of 2 cm × 1 cm × 1 cm) that had been naturally fermented for three days at 28 ± 2 °C, with 75 ± 5% relative humidity and a photoperiod of 16:8 h (L/D).

### 2.2. Total RNA Extraction, Sequencing, and CYP450 Gene Identification

Whole females (3-day-old virgins) were exposed to β-cypermethrin (Gongcheng Bio-tech Co., Ltd., Nantong, China) for 0 or 1 h and then underwent RNA sequencing using three replicates. RNA was extracted using TRIzol^®^ Reagent (Takara, Japan) according to the manufacturer’s protocol. Libraries were constructed using the NEB Next Ultra RNA Library Prep Kit and finally sent to Beijing Annoroad Gene Technology Co., Ltd. (Beijing, China) for transcriptome sequencing on the Illumina MiSeq platform with 150 bp paired-end sequencing. Raw reads are available in the Genome Sequence Archive under the accession CRA008976 and project PRJCA013354 that are publicly accessible at https://ngdc.cncb.ac.cn/gsa (accessed on 20 November 2022). Transcriptome assembly was accomplished using Trinity software v2.3.1 [22]. All assembly transcripts were annotated using BLASTX to search the highest sequences similarity against the NCBI non-redundant (NR) protein database with a cutoff e-value < 10^−5^. Transcript abundance was calculated by the RPKM (reads per kilobase per million mapped reads) method [23], and differential expression analysis was performed using DESeq2 software [24]. The screening criteria for differently expressed genes (DEGs) were corrected *p*-values < 0.05 and *|*log2(foldchange)*|* > 1. Functional annotation and GO enrichment analyses were carried out using the R and Cluster Profiler package [25,26] to categorize the DEGs into biological processes (BPs), molecular functions (MFs), and cellular components (CCs). The CYP450 genes were searched and analyzed from the transcriptome database that provided sequence information.

### 2.3. Cloning of the Upregulated CYP450 Genes

Total RNA was isolated as described above, and the cDNA was synthesized from 500 ng of total RNA using the Prime script™ RT Reagent kit (Takara, Japan) according to the manufacturer’s instructions. Five pairs of specific primers (Appendix A) were used to amplify the full-length sequences with the following cycling parameters: 95 °C for 3 min, followed by 32 cycles at 94 °C for 25 s, 55 °C for 30 s, and 72 °C for 1 min, with a final step of 72 °C for 5 min. The purified PCR products were subcloned into the pClone007 Cloning Vector (Qingke Biotech Co., Ltd. Beijing, China) and then sequenced by Qingke Biotech Co., Ltd. (Chengdu, China).

### 2.4. Bioinformatics Analysis

The molecular weight and theoretical isoelectric point (pI) of the predicted CYP4D2, CYP49A1, and CYP28D2 proteins were determined using the Compute PI/Mw tool (https://web.expasy.org/protparam/, accessed on 28 March 2021). The amino acid sequences were aligned using Clustal X1.81 and edited using DNAMAN 6.0 with related amino acid sequences, which were obtained by protein BLAST searching through the NCBI homepage (available online: https://blast.ncbi.nlm.nih.gov/Blast.cgi, accessed on 28 March 2021). The protein domains were identified by searching the Pfam database (http://pfam.xfam.org/, accessed on 28 March 2021), and the conserved motifs were identified using MEME online software (http://meme-suite.org/tools/meme, accessed on 28 March 2021). The phylogenetic tree was constructed using MEGA software (V11.0) and the neighbor-joining method with 1000 bootstrap replications.

### 2.5. Expression Profiles of Three CYP450 Genes

Three tissues, namely, the MG, FB, and MTs, were dissected (Appendix A) from 3-day-old virgin female *P. okadai* (*n* = 20) that were treated with 0.166 mg/L of β-cypermethrin for 0, 0.5, 1, 2, 3, and 4 h. Total RNA was extracted, and cDNA templates synthesized following the procedure described in Section 2.3. Three replicates were analyzed, with the 0 h treatment used as the control group. Real-time quantitative PCR (RT-qPCR) was performed on the Applied Biosystems 7500 system (Applied Biosystems, Waltham, MA, USA) with the SYBR Green qPCR Master Mix (Takara, Japan). *β-Tubulin* was used as a reference gene [27], and three technical replicates were performed for the RT-qPCR experiments. Amplification was conducted as follows: 95 °C for 5 min, followed by 39 cycles at 95 °C for 15 s and 60 °C for 40 s. A melting curve analysis was then performed to assess the specificity of each reaction. The 2^−∆∆Ct^ method [28] was used to compare the expression levels of the three targeted genes.

### 2.6. Functional Expression of Pocyp4d2 in S2

The open reading frame (ORF) of *Pocyp4d2* gene was codon-optimized for mammalian expression and synthesized by Genscript (Gongcheng Bio-tech Co., Ltd., Nantong, China) including a C-terminal 6 × His tag, *BamH* I and *EcoR* I restriction endonuclease recognition sites. The synthesized *Pocyp4d2* gene was cloned into the pFastbac1-EGFP vector (Beyotime, Co., Ltd., Shanghai, China), generating the pFastBac1-*Pocyp4d2* plasmid. The recombinant pFastBac1-*Pocyp4d2* plasmid was transformed into *Escherichia coli* Top 10 competent cells for the amplification of the recombinant plasmid, and then positive clones were selected for further PCR and sequencing to verify the authenticity of the target gene. The sequencing-verified plasmids were then transformed into *E. coli* cells (Gongcheng Bio-tech Co., Ltd., Nantong, China) to obtain a recombinant bacmid. *Drosophila melanogaster* Schneider 2 (S2) cells were transfected with the recombinant bacmid using Cellfectin II reagent (Promega, Madison, WI, USA) according to the manufacturer’s instructions, and then the infected cells were harvested and lysed using the lysis buffer provided with the Membrane and Cytosol Protein Extraction Kit (Beyotime, Co., Ltd., Shanghai, China). The purified his-PoCYP4D2 was assessed by electrophoresis on a 10% SDS-polyacrylamide gel (SDS-PAGE), and a 47.5 kDa band was visualized by silver staining and Western blot (WB). In addition, immunofluorescence analysis was performed on the his-PoCYP4D2 with 2-(4-amidinophenyl)-6-indolecarbamidine dihydrochloride (DAPI) and fluorescein isothiocyanate isomer I (FITC) (Beyotime, Co., Ltd., Shanghai, China). To investigate the effect of PoCYP4D2 on β-cypermethrin-inhibited S2 cells using the CCK8 assay, the cells were divided into the following groups: a blank group (S2 cells), a β-cypermethrin group, a Bacmind-PoCYP4D2 + β-cypermethrin group, and a Bacmind-PoCYP4D2 group.

### 2.7. Functional Analysis of Pocyp4d2 by RNAi

To further assess whether the *Pocyp4d2* genes are involved in β-cypermethrin detoxifications in *P. okadai*, RNAi was performed by injecting small interfering RNA (siRNA) into virgin *P. okadai* females. The full-length cDNAs of *Pocyp4d2* were used as templates for siRNA synthesis (Appendix A). All siRNA oligonucleotides were synthesized by Baiqi Biotech Co., Ltd. (Wuhan, China). Each individual was injected with 0.5 μL of siRNA (1 μg/μL of siRNA) for the *Pocyp4d2* gene into the back between the wings using a micro-injector (longerpump TJ-1A, Beijing, China) under a microscope (Olympus, Tokyo,, Japan). Sixty *P. okadai* insects that had received the siRNA injections were used to characterize the transcript levels of the *Pocyp4d2* gene, which were measured by RT-qPCR at 24 and 48 h, and the 5′-6-carboxyfluorescein (FAM)-injected insects were taken as the control group. The experiment was repeated three times, with three technological replicates each time.

To evaluate the role of the *Pocyp4d2* gene in the detoxification of β-cypermethrin, 500 μL of insecticide β-cypermethrin (0.166 mg/mL), dissolved in acetone, was topically applied onto the bottle and placed in ventilated cabinets overnight. The mortality of the insects was tested using the drug-film method at 24 h after *Pocyp4d2*-siRNA and FAM had been injected into the back of *P. okadai*. Each siRNA-injected group and the control group contained 20 individuals, and the experiments were performed four times. The mortality of the insects was recorded 3 h after the insecticide treatments.

### 2.8. Data Analysis

The mRNA transcription levels of the CYP450 genes in different developmental stages and tissues were analyzed by one-way ANOVA. The Duncan test was used for multiple comparisons, followed by the Tukey’s honestly significant difference test. All the data are shown as the mean ± SE, and differences were considered statistically significant at *p ≤* 0.05; the statistical analyses were carried out using GraphPad prism8.0 or SPSS18.0.

## 3. Results

### 3.1. Identification of Upregulated CYP450 Genes by RNA Sequencing in P. okadai

The transcriptome differential expression analysis showed that there were 16,755 DEGs in *P. okadai* exposed to β-cypermethrin for 0 and 1 h, among which 7640 DEGs were more highly expressed in the 1 h group, and 9115 DEGs were higher in the 0 h group (Appendix A). For better understanding the differences in regulation between the two groups, we mapped all the transcripts to GO pathways to identify the pathways that were significantly enhanced. The DEGs with higher expression levels were enriched in the pathways cellular processes and biological regulation in BP, cell part and membrane part in CC, and binding and catalytic activity in MF (Appendix A). The transcriptome analysis revealed the expression of 21 different CYP450 genes. Compared with the 0 h group, there were 5 upregulated and 16 downregulated CYP450 genes (Appendix A).

### 3.2. Cloning and Bioinformatic Analysis of Pocyp4d2, Pocyp49a1, and Pocyp28d2

Using *P. okadai* cDNA, the full-length coding sequences of *Pocyp4d2* (GenBank Accession No. OP555111), *Pocyp49a1* (GenBank Accession No. OP555112), and *Pocyp28d2* (GenBank Accession No. OP555113) were PCR-cloned and confirmed by sequencing, except for *Pocyp28d1*and *Pocyp28d3* (Appendix A). These three genes belong to the CYP4 family, the CYP Mito family, and the CYP28 family, respectively. PoCYP4D2 encodes a protein of 419 amino acids with a predicted mass and a pI of 48.2 kDa and 6.6, respectively; PoCYP49A1 encodes a protein of 407 amino acids with a predicted mass and a pI of 46.75 kDa and 9.17, respectively; PoCYP28D2 encodes a protein of 300 amino acids with a predicted mass and a pI of 33.98 kDa and 5.05, respectively. MEME software analysis showed that each sequence contains two or three distinct characteristics of the CYP450 protein sequences, PoCYP4D2 has one C-helix motif and two K-helix motifs, PoCYP49A1 has one C-helix motif, three K-helix motifs, and one heme-iron ligand signature motif, and PoCYP28D2 has two K-helix motif and one heme-iron ligand signature motif (Appendix A). A phylogenetic tree was constructed using the neighbor-joining method to investigate the evolutionary relationships among these three CYP450 genes from *P. okadai* and other insects (Figure 1). The phylogenetic analysis results showed that PoCYP49A1 and PoCYP28D2 are closely related to the genes of *Drosophila* spp., while PoCYP4D2 is closely related to genes of *Lucillia euprina* and *Musca domestica*. The online sequence similarity analysis (http://www.ncbi.nlm.nih.gov/BLAST/, accessed on 29 March 2021) showed that PoCYP4D2 shares 62.44% similarity with a homolog from *Lucilia cuprina*. The amino acid similarity to most *Drosophila* genes was found to range from 50% to 60%. PoCYP49A1 showed 82.93% similarity with the *Drosophila miranda* gene, and the *Pocyp28d2* gene showed 64.67% similarity with the *Drosophila serrata* gene.

### 3.3. Expression Profiles of Pocyp4d2, Pocyp49a1, and Pocyp28d2 following β-Cypermethrin Exposure

The expression profiles of the CYP450 target genes in the MG, FB, and MTs were analyzed under β-cypermethrin stress after exposure for 0, 0.5, 1, 2, 3, and 4 h; we considered the 0 h stage as the control. *Pocyp4d2* expression in the FB and MG showed a maximum increase after 1 h of exposure, and the expression in the MTs started to increase at 0.5 h and peaked at 4 h after β-cypermethrin exposure (Figure 2A). *Pocyp4d2* expression was the highest in the MG at 1 h (*p* < 0.05) and was likewise present in the FB and MTs. *Pocyp49a1* expression in the FB and MG showed a maximum increase after 1 h of exposure, and the expression in the MTs peaked at 4 h after β-cypermethrin exposure (Figure 2B). *Pocyp49a1* expression was highest in the FB at 1 h and was similarly present in the MG and MTs. *Pocyp28d2* expression in the FB and MG showed a maximum increase after 1 h of exposure, and the expression in the MTs began at 0.5 h and peaked at 4 h after β-cypermethrin exposure (Figure 2C). *Pocyp28d2* expression was highest in the FB at 1 h and was also present in the MG and MTs. The expression levels of these three CYP450 genes in the MG were significantly increased, with values that were 2.82-fold (*Pocyp4d2*), 2.62-fold (*Pocyp49a1*), and 1.77-fold (*Pocyp28d2*) higher following β-cypermethrin exposure for 1 h compared with the levels at 0 h (Figure 2D), while *Pocyp4d2* was the most upregulated gene and reached a peak at 1 h (*p* < 0.05). Therefore, *Pocyp4d2* was selected as the candidate gene for research into its detoxification function.

### 3.4. Functional Expression of Pocyp4d2 in S2 Cells

#### 3.4.1. Identification of the Recombinant Plasmid pFastBac1-*Pocyp4d2* and the PoCYP4D2 Protein

The positive clones were collected, cleaved with *Bam*HI and *Eco*RI, and detected with PCR amplification (Figure 3A). The recombinant protein PoCYP4D2 was identified by SDS-PAGE and Western blot, was about 47.5 kilodaltons, expressed in the S2 cells lysis pellet rather than in the cell lysis supernatant, as expected (Figure 3B,C), and was distributed in the cytoplasm, as detected by immunofluorescence (Figure 3D).

#### 3.4.2. Metabolic Detoxification Function of the Recombinant Protein PoCYP4D2 in S2 Cells

To reveal the role of PoCYP4D2 in β-cypermethrin tolerance, the recombinant protein PoCYP4D2 was transferred into S2 cells. The viability of the S2 cells decreased significantly in both the β-cypermethrin group and the Bacmind-PoCYP4D2 + β-cypermethrin group compared with the blank group (*p* < 0.05 or *p* < 0.01), and the viability was significantly lower for cells in the blank group than in the Bacmind-PoCYP4D2 group (*p* < 0.01); however, the proliferation of the cells in the β-cypermethrin group was significantly lower than that of the cells in the Bacmind-PoCYP4D2 + β-cypermethrin group (*p* < 0.01) (Figure 4). Therefore, the recombinant protein PoCYP4D2 significantly decreased the inhibition of S2 cells by the β-cypermethrin treatment, as detected by the CCK8 assay.

### 3.5. Functional Analysis of Pocyp4d2 by siRNA

After siRNA injection, the mRNA transcription levels of *Pocyp4d2* in *P. okadai* in the FB, MG, and MTs were reduced at 24, 48, and 72 h, respectively. The mRNA transcription levels of *Pocyp4d2* in the FB were reduced to 61.78%, 67.03%, and 78.32%; the mRNA transcription levels of *Pocyp4d2* in MTs were reduced to 65.99%, 44.03%, and 73.26%; and the mRNA expression levels of *Pocyp4d2* in MG were suppressed to 58.77%, 37.28%, and 60.73%, with the target gene being expressed at the lowest levels in the MG at 48 h (Figure 5).

The mortality of the *P. okadai* insects increased from 6.25% to 15.0% after *Pocyp4d2* was silenced at 3 h under β-cypermethrin stress; however, the mortality of the *P. okadai* insects further increased from 15.0% to 27.5% after 6 h (Figure 6); the mortality of the knockdown insects increased significantly with cumulative increases in the exposure time.

## 4. Discussion

*T. callipaeda* infection in *P. okadai* chiefly occurs in areas where wildlife is found, with *P. okadai* acting as its intermediate host, the only confirmed vector in China at present [29,30]. Many cases in which people were reportedly infected with *T. callipaeda* involved exposure to domestic animals (dogs and cows) or patients residing near pear plantations with regular pesticide spraying. We do not yet know whether the development of *P. okadai* resistance increases the risk of *T. callipaeda* infection. Still, the increasing resistance of insects with enhanced CYP450 gene expression may lead to increased difficulties in pest control, as clearly shown in numerous studies [16,18]. The CYP4 and CYP6 clans are known to play an essential role in xenobiotic metabolism and insecticide resistance and are closely associated with resistance to pyrethroid insecticides and many other insecticides [31,32]. We focused on the expression changes of the CYP450 genes in the midgut and other related insect tissues under β-cypermethrin pesticide exposure, since the pesticides function chiefly through ingestion and contact. In this study, *Pocyp4d2* was the most upregulated gene and reached a peak in the midgut at 1 h, according to our spatiotemporal expression profile analysis. Therefore, we selected *Pocyp4d2* as the candidate gene for detoxification function research in this study.

To provide an initial assessment regarding the functions of the *Pocyp4d2* gene under β-cypermethrin stress, we examined the expression patterns of the *Pocyp4d2* gene in various tissues and different action stages of 3-day-old virgin females using RT-qPCR. The *Pocyp4d2* gene was found mainly distributed in the midgut, fat body, and Malpighian tubules of *P. okadai* and expressed at the highest levels in the midgut at 1 h. The midgut, fat body, and Malpighian tubules have frequently been recognized as the principal organs of acute insecticide poisoning following insecticide metabolism [33]. We found that *Pocyp4d2* was knocked down to 37.28% in the midgut minimum, to 67.03% in the fat bodies, and to 44.03% in the Malpighian tubules by *Pocyp4d2*-siRNA interference, mainly at 48 h. At the same time, remarkably, the mortality of *P. okadai* (after interference with *Pocyp4d2*-siRNA) increased at 3 h and 6 h with respect to that of the control with the same concentration of β-cypermethrin, which suggests that the *Pocyp4d2* gene may be involved in the metabolism of β-cypermethrin in *P. okadai*. Wang et al. found that enhanced *cyp321b1* gene expression was detected predominantly in *Tobacco cutworms* and could participate in the metabolic detoxification of β-cypermethrin in the midgut; additionally, the mortality of the insects was increased after *cyp321b1* was silenced [34]. The *cyp6hl1* and *cyp6hn1* genes were predominantly distributed in the *Locusta migratoria* fat body, and their knockdown significantly increased nymph mortality following exposure to cypermethrin or fenvalerate [35]. *D*. *melanogaster* dietary exposure to permethrin and *cyp4e3* knockdown caused a significant elevation of oxidative stress-associated markers in the Malpighian tubules, which included lipid peroxidation based on the production of 4-hydroxynonenal; these findings have increased our understanding of the molecular mechanisms of permethrin detoxification in the Malpighian tubules [36]. These observations indicate that the fat body, midgut, and Malpighian tubules are essential tissues for detoxification [37,38]. In particular, the expression of *Pocyp4d2* was higher in the midgut than in other tissues. The high expression of *Pocyp4d2* in the midgut may be related to the detoxification of β-cypermethrin acting in the digestive system, but it may also involve the synergistic interaction of the fat body and Malpighian tubules.

Furthermore, the CCK8 assay showed that the viability of S2 cells was significantly higher in the Bacmind-PoCYP4D2 group than in the blank group, which indicated that Bacmind-PoCYP4D2 may be both associated with β-cypermethrin metabolism and involved in cell proliferation. These studies showed that some of the P450 enzymes could participate in the regulation of insect growth and development through metabolic detoxification [39]. *Hyphantria cunea* moth larvae were found to be able to upregulate P450 enzyme activity and metabolize and detoxify the flavonoids contained in *Begonia* leaves [40] and the phenolic secondary metabolite chlorogenic acid [41], which could affect their feeding strategy and allow them to adapt to different hosts for growth and development. Therefore, further research on the metabolism and detoxification of the *P. okadai* P450 enzyme and their interaction with growth and development is needed.

However, studies on the resistance of *Pocyp4d2* in *P. okadai* have not been reported; the enhanced *cyp4d2* expression found in *D. melanogaster* [42] and *Cydia pomonella* [43] supports its role in deltamethrin insecticide metabolism. A decrease in resistance to permethrin of mosquito larvae was strongly associated with the knockdown of the target P450 genes, namely, to *cyp4d42v1* knockdown to 0.35, *cyp6p14* to 0.55, and *cyp49al1* to 0.55 of the original levels, by RNAi in *Culex mosquitoes* [44]. The expression levels of the *cyp4m51* and *cyp6ab56* genes were enhanced by deltamethrin in cabbage moth larvae (*Mamestra brasicae Linnaeus*), but the knockdown of the targeted CYP450 genes was found to reduce deltamethrin sensitivity, which caused the mortality of the larvae to increase by 11.4% and 21.6%, respectively [45]. Hence, pyrethroid resistance was found to be closely related to the overexpression of the CYP4, CYP6, and CYP9 family genes in insect species by many researchers worldwide [46]. In addition, we found that the expression levels of the *Pocyp49a1* and *Pocyp28d2* genes were upregulated in metabolic detoxification-related tissues following β-cypermethrin treatment, which implies that there are other CYP450 genes involved in detoxification besides *Pocyp4d2*, which we focused on in this study. The Mito CYP family gene *cyp49a1* was found to be upregulated following phoxim treatment in the fat body of silkworm (*Bombyx mori*). In contrast, the CYP28 family has been implicated in the metabolism of insecticides and toxic natural plant compounds in insects [47,48]. Multiple CYP450 genes may be involved in the metabolic detoxification of β-cypermethrin in *P. okadai*.

## 5. Conclusions

In conclusion, we found that *Pocyp4d2* decreased the inhibition of β-cypermethrin following its overexpression in S2 cells and that a subsequent *Pocyp4d2*-siRNA injection significantly increased the mortality of *P. okadai* after exposure to the same concentration of β-cypermethrin. These results imply that *Pocyp4d2* may be an essential gene in the metabolism of β-cypermethrin in *P. okadai* and that controlling *P. okadai* through *Pocyp4d2* inhibition/interference may be a reasonable strategy.

## Figures and Tables

**Figure 1 genes-13-02338-f001:**
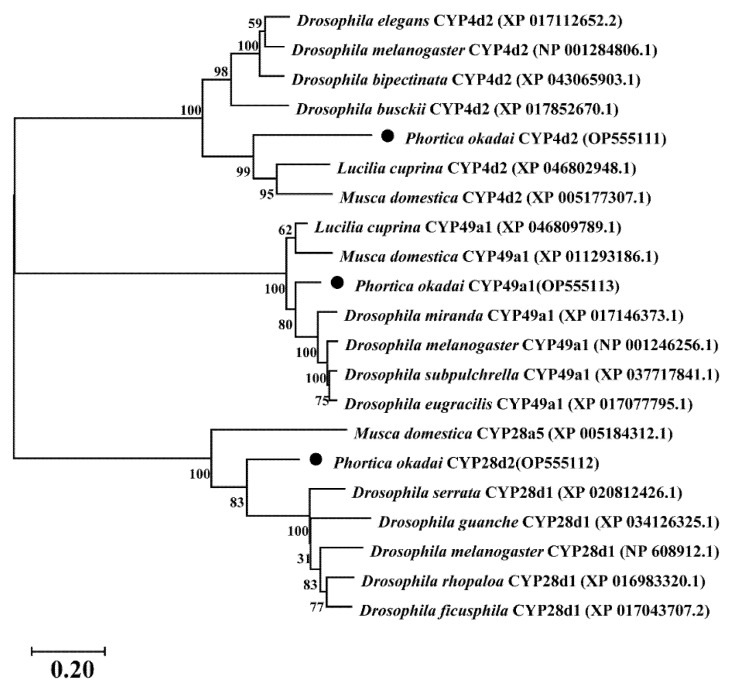
Phylogenetic analysis of CYP4D2, CYP49a1, and CYP28D2 of *Phortica okadai* and related P450s. The scale bar indicates 0.20 amino acid substitutions per site and the branch length represents genetic distance and the value on the branch is the support rate. The three *P. okadai* CYP450s are indicated by “●”.

**Figure 2 genes-13-02338-f002:**
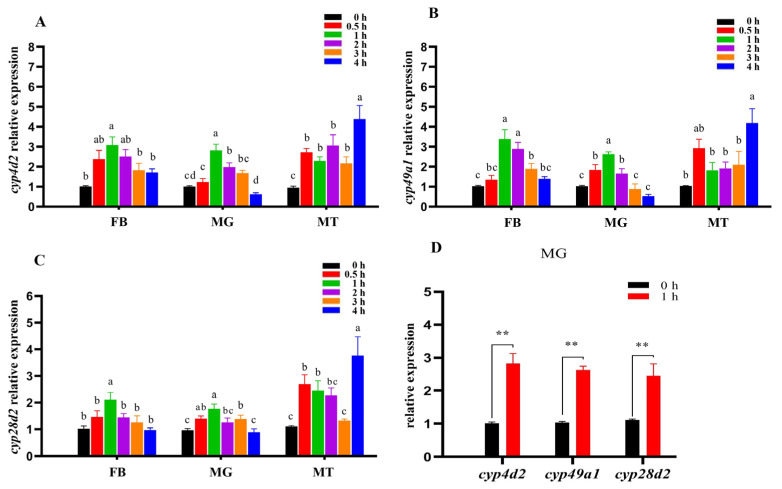
Spatiotemporal expression profiles of *Pocyp4d2*, *Pocyp49a1*, and *Pocyp28d2* in different tissues of *P. okadai* after β-cypermethrin treatment ($\bar{X}$ ± s, *n* = 4). The spatiotemporal expression profile of (**A**) *cyp4d2*, (**B**) *cyp49a1*, and (**C**) *cyp28d2*. (**D**) Relative expression levels of *Pocyp4d2*, *Pocyp49a1*, and *Pocyp28d2* in MG. Note: Different letters above bars indicate significant differences between groups (Duncan’s multiple range comparison; ** *p* < 0.01).

**Figure 3 genes-13-02338-f003:**
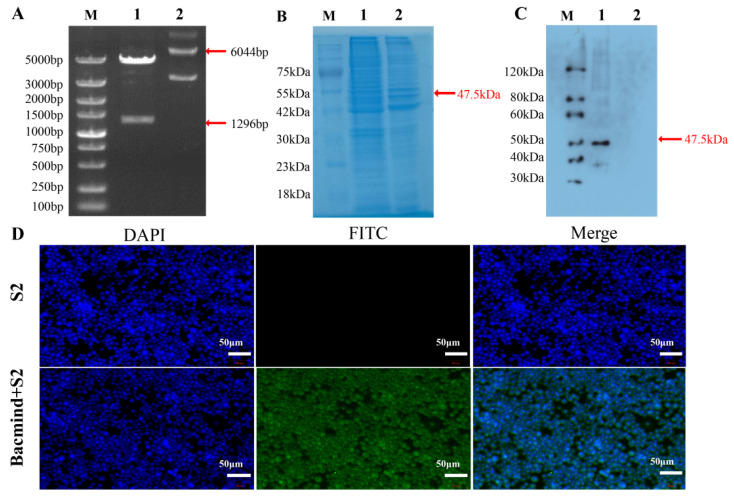
Identification of the recombinant plasmid pFastBac1*-Pocyp4d2* and recombinant protein PoCYP42. (**A**) Restriction enzyme analysis of the recombinant plasmid pFastBac1*-Pocyp4d2*; M: 5000 bp DNA ladder; 1: *Bam*HI and *Eco*R I digestion of the recombinant plasmid pFastBac1*-Pocyp4d2*; 2: pFastBac1*-Pocyp4d2* recombinant plasmid. (**B**) SDS-PAGE analysis of the recombinant protein PoCYP4D2; M: 75 kDa protein ladder 1: lysed cells’ precipitate; 2: lysed cells’ supernatant. (**C**) Western blot analysis of the recombinant protein PoCYP4D2; M: 120 kDa protein ladder 1: lysed cells’ supernatant; 2: lysed cells’ precipitate. (**D**) Immunofluorescence identification of PoCYP4D2 recombinant protein in S2 cells after 72 h (200×).

**Figure 4 genes-13-02338-f004:**
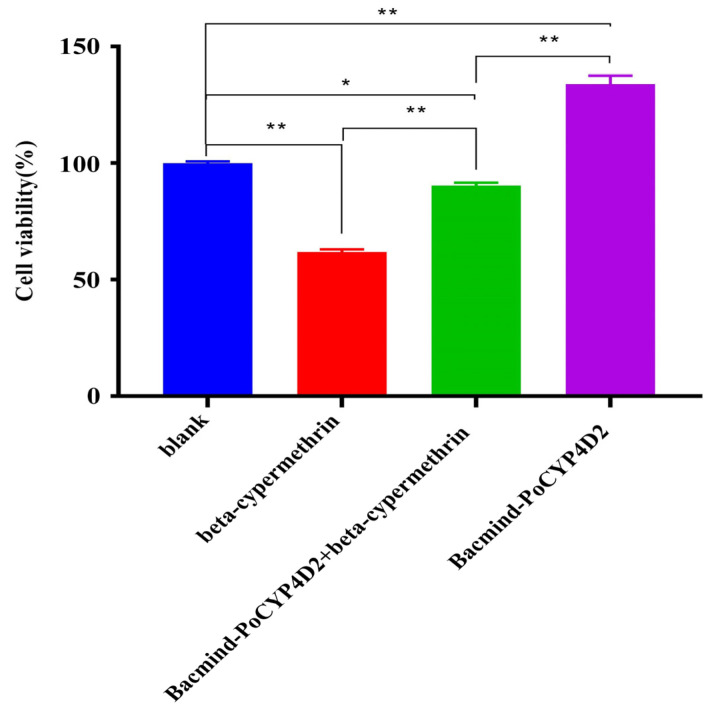
Effects of bacmind transfection on the inhibition of S2 cells by the β-cypermethrin treatment ($\bar{X}$ ± s, *n* = 3) Note: Different colors means different groups, * *p* < 0.05, ** *p* < 0.01.

**Figure 5 genes-13-02338-f005:**
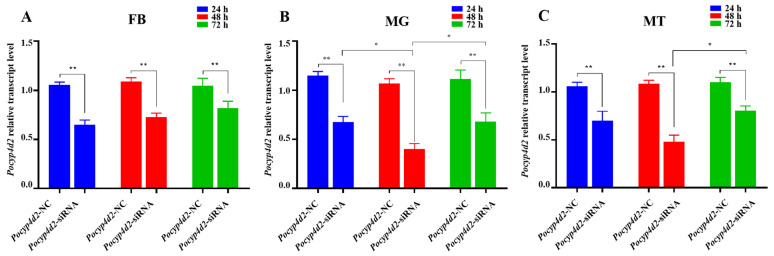
Changes in *Pocyp4d2* relative expression in different tissues of *P. okadai* after *Pocyp4d2*-siRNA interference as detected by RT-qPCR ($\bar{X}$ ± s, *n* = 3). Effects of *Pocyp4d2* knockdown in the (**A**) fat body, (**B**) midgut, and (**C**) Malpighian tubules. Note: * *p* < 0.05, ** *p* < 0.01.

**Figure 6 genes-13-02338-f006:**
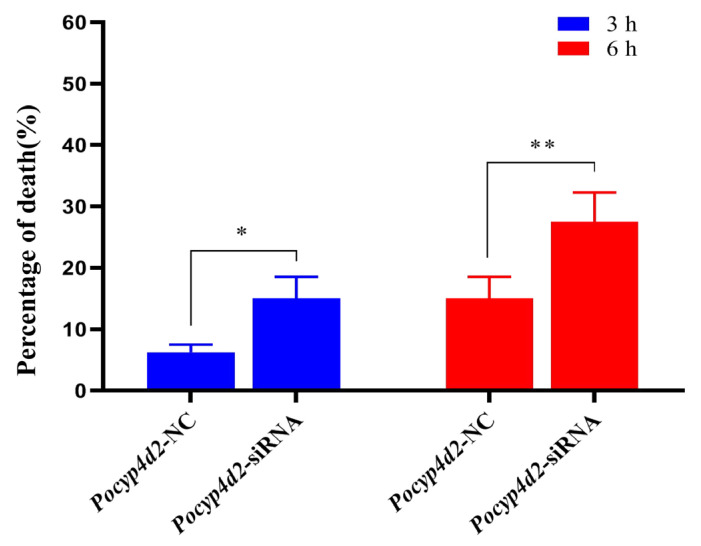
Effects on the mortality of *P. okadai* under β-cypermethrin stress exerted by *Pocyp4d2*-siRNA interference ($\bar{X}$ ± s, *n* = 4). Note: * *p* < 0.05, ** *p* < 0.01.

## Data Availability

All the supporting data and protocols have been provided within the article or in the Appendix A.

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
