# Peer review of "Identification and Functional Characterization of CYP4D2 Putatively Associated with β-Cypermethrin Detoxification in Phortica okadai"

_genes, 2022, doi:10.3390/genes13122338_

Round 1
Reviewer 1 Report
Comments to authors:
The manuscript by Wang et al. identified Beta-Cypermethrin detoxifying cytochrome P450 isoform in Phortica okadai. The authors have performed RNA-sequencing to identify the genes that are differentially regulated in response to Beta-Cypermethrin insecticide treatment. Then they performed siRNA-mediated knockdown and overexpression studies in the target insect and Drosophila-derived S2 cell line, respectively to understand the role of identified CYP gene in the detoxification of Beta-Cypermethrin. The study is intriguing, though in its current state it suffers from many deficiencies which are listed below.
Additionally, there are numerous grammatical and spelling errors throughout the paper. Some sentences are very difficult to understand. Proofreading this manuscript by a native English speaker may improve the readability.
Abstract:
Line No. 20: What are 0 and 1 hand? Is it 0 and 1 hour? Please correct it.
Line No. 23-24: This sentence is not clear, rewrite.
Introduction:
Line No. 73-74: This sentence is not clear, rewrite.
Materials and Methods
Line No: 90-92: The authors need to provide details about how they assembled the transcriptome and performed RNA-sequencing analysis.
Line No: 91-92: Ideally, 2-fold differences in the expression are assumed to be biologically relevant. I am concerned that 1-fold differences may not represent a real biological variation. I suggest the authors apply ≥2-fold criteria to see whether the predicted CYP genes are significantly altered or not in Beta-Cypermethrin treatment compared to the control.
Line No. 92: The authors need to provide the full form of the abbreviation “padj” when they first introduced it. Please check and correct other abbreviations throughout the manuscript.
Line No.118: How many insects were used for dissecting and extracting MG, FB, and MTs?
Line No. 135: The authors need to describe how they expressed the selected CYP in S2 cells.
Results:
Lino No. 180: I wonder what the reason in selecting only 3 out of 5 upregulated genes for cloning and further studies.
Line No. 195: The authors need to provide what is the importance of C- and K-helixes.
Line No. 225-229: There is not much difference between Pocyp4d2 and Pocyp49a1 genes expression levels. I wonder if all three CYP genes may be functioning in the detoxification of Beta-Cypermethrin. The authors need to justify why they only focused on Pocyp4d2.
Line No. 240: I do not understand why the authors did PCR if they already digest with respective restriction enzymes to confirm the release of the insert.
Line No. 242: How did you perform lysis of S2 cells?
Discussion:
Line No. 308-312: Pocyp4d2 knocking down levels alone does not represent its role in the detoxification of Beta-Cypermethrin. Knocking down of this CYP gene increases the susceptibility of P. okadai to Beta-Cypermethrin. Modify this sentence.
Line No. 368: To support this statement, there is no empirical data in this manuscript. I suggest the authors express other overexpressed CYP genes in S2 cells along with Pocyp4d2 to check synergistic effects of multiple CYPs.
Data availability: RNA-sequencing data should be deposited into the public domain like NCBI for everyone's access.
Reviewer 2 Report
Dear authors
Please find the attached file.

Reviewer 3 Report
General comments for authors
The manuscript titled “Identification and Functional Characterization of cyp4d2 Putatively Associated with Beta-Cypermethrin Detoxification in Phortica okadai)” represent a significant contribution to scholarly research and is good for publication but needs major revisions.
1. Abstract: The information provided in the abstract did not reflect the title of the study- For instance the identified gene associated with Beta cypermethrin detoxification in P. okadai was not clearly stated – thus the abstract should be revised to reflect the major findings of the study
2. In introduction; The clear objectives of the study should be stated towards the end of introduction .
3. Materials and methods should be more explicit. Some sub titles are not well captioned and should be addressed.
4. Results. Presentation in result section is poor , Authors repeated the procedures used which is supposed to presented in materials and methods in results section instead of discussing the results right away. The whole results sections should be properly rewritten to capture only the results of the study for clarity
5. Discussion needs improvement
6. Other comments are found in the reviewed text.
Thank you

Round 2
Reviewer 1 Report
The revised manuscript by Wang et al. sufficiently addressed my comments raised during the initial review. The current version looks better, though I have a few minor suggestions/corrections that need to be addressed before accepting this manuscript.
Minor corrections:
Line 21: Delete “and then, gene cloning of the five up-regulated CYP450 genes was further performed”.
Line 27: Write the full form of RNAi and siRNA.
Line 106: Is it false discovery rate (FDR) corrected P values?
Fig. 3A. gel picture shows the release of the insert by respective restriction enzymes digestion. If the authors reconfirmed the target gene insertion by PCR, include this data in the manuscript.
Reviewer 3 Report
Authors have improved the quality of the manuscript on this revised version